# Concurrent Disorder Management Guidelines. Systematic Review

**DOI:** 10.3390/jcm9082406

**Published:** 2020-07-28

**Authors:** Syune Hakobyan, Sara Vazirian, Stephen Lee-Cheong, Michael Krausz, William G. Honer, Christian G. Schutz

**Affiliations:** 1Department of Psychiatry, The University of British Columbia, Vancouver, BC V6T 1Z4, Canada; Sara.Vazirian@alumni.ubc.ca (S.V.); Michael.Krausz@ubc.ca (M.K.); William.Honer@ubc.ca (W.G.H.); Christian.Schutz@ubc.ca (C.G.S.); 2Department of Public Health, King’s College London, London WC2R 2LS, UK; stephenleecheong@gmail.com

**Keywords:** concurrent disorder, co-occurring disorder, dual diagnosis, dual pathology, addiction comorbidity, comorbid substance abuse, comorbid illicit use, comorbid addiction, comorbid mental illness, coexisting mental illness

## Abstract

Concurrent disorder refers to a diverse set of combinations of substance use disorders and mental disorders simultaneously in need of treatment. Concurrent disorders are underdiagnosed, undertreated, and more complex to manage, practicing the best recommendations can support better outcomes. The purpose of this work is to systematically assess the quality of the current concurrent disorders’ clinical recommendation management guidelines. Literature searches were performed by two independent authors in electronic databases, web, and gray literature. The inclusion criteria were English language clinical management guidelines for adult concurrent disorders between 2000 and 2020. The initial search resulted in 8841 hits. A total of 24 guidelines were identified and assessed with the standardized guidelines assessment tool: AGREE II (Appraisal of Guidelines for Research and Evaluation). Most guidelines had acceptable standards, however, only the NICE guidelines had all detailed information on all AGREE II Domains. Guidelines generally supported combinations of treatments for individual disorders with a very small evidence base for concurrent disorders, and they provided little recommendation for further structuring of the field, such as level of complexity or staging, or evaluating different models of treatment integration.

## 1. Introduction

Concurrent disorder (also called dual diagnosis, co-occurring disorder, comorbidity) refers to a specific form of multimorbidity within the area of mental health, where at least one substance use disorder and at least one non-substance-bound mental disorder is simultaneously in need of treatment. The World Health Organization (WHO) defined dual diagnosis as the co-occurrence of a psychoactive substance use disorder and another psychiatric disorder in the same individual [1]. The European Monitoring Centre for Drugs and Drug Addiction (EMCDDA) defined comorbidity/dual diagnosis as the temporal coexistence of two or more psychiatric disorders as defined by the International Classification of Diseases, one of which is problematic substance use. To describe the co-occurring mental health and substance use disorders, other terms have been used as well. The Canadian accepted term is “concurrent disorder” [2]. The US-American accepted term is “co-occurring disorders” [3]. The term “comorbidity” is used in Australia; however, recently more descriptive terms have been used: “coexisting mental health and substance use disorders” or “coinciding mental illness and substance abuse”. The term “coexisting problems” is used in New Zealand. “Chemically affected Mental Illness” (CAMI), “Mental Illness Chemically Affected” (MICA), “Substance Affected Mentally Ill” (SAMI), “Mental Illness Substance Affected” (MISA), “Mental Illness Substance Use Disorder” (MISUD), and “Individuals with Co-Occurring Psychiatric and Substance Use Disorders” (ICOPSD) are other terms used to describe the same condition [4]. The term “dual diagnosis” is frequently used in the United Kingdom, Australia, Spain, and Spanish speaking countries. Adding to the confusion, the term “dual diagnosis” is applied for concurrent intellectual or developmental disorders with mental health disorders in Canada. For the purpose of this work the intellectual and developmental disabilities with mental health concerns that are considered as a “dual diagnosis” or “concurrent disorder”, will not be considered or discussed.

In mental health, the focus of research and guidelines has been on individual disorders, despite concurrent disorders being common and seemingly increasing [5]. Substance use disorders and non-substance-related mental disorders are frequently chronic, requiring long-term care. Greater severity of a single psychiatric disorder increases the risk of developing concurrent disorders. This also means that in general the frequency of comorbidity increases from population-based studies, to outpatient studies, to inpatient studies. In population-based studies, approximately one-fourth of people with anxiety or major depressive disorders are expected to have an overlapping substance use disorder in their lifetime [6,7]. Similarly, half of the people with bipolar disorder or schizophrenia will experience a substance use disorder [8]. Studies generally exclude tobacco dependence, otherwise the numbers would be substantially higher.

People with concurrent disorders tend to be underdiagnosed and undertreated, whilst experiencing a high burden of morbidity and mortality. There are big gaps between the need for substance use disorders, mental disorders treatment, and delivered services. Unmet need for treatment is more for substance use disorders. Psychiatrists are often uninvolved with the management of substance use disorders, and general or addiction physicians treating substance use disorders do not necessarily diagnose psychiatric disorders. The treatment of psychiatric disorders and substance use disorders is separated in many countries, with different treatment traditions, separate organizations within the healthcare system, separate treatment providers, and separate funding. Individuals with concurrent disorders are not only more complex to diagnose and treat, but they are also at higher risk of additional multimorbidity, becoming socially marginalized, entangled with the legal system, and subject to stigma [9]. Both mortality and morbidity are increased in those with concurrent disorders. The main causes are premature drug-related death [10] and increased risk of suicide [11,12]. Increased utilization of healthcare services has been demonstrated, despite the demonstrated treatment gap. For example, in a Canadian cohort study, individuals with concurrent disorders had significantly higher odds of Emergency Department use (Adjusted odds ratio [AOR] D 1.71; 95% confidence interval [CI]), 1.4–2.11, hospitalization (AOR D 1.45; 95% CI, 1.16–1.81), and primary care visits (AOR D 1.34; 95% CI, 1.05–1.71) than those with either substance use disorder or non-substance-related mental disorders [13].

The mechanisms of development of concurrent disorders are complex, however, frequently both conditions share neurological pathways, overlapping underlying genetic risk factors, as well as common “environmental” risk factors. People with concurrent disorders are frequently part of a highly vulnerable population—with multiple biological, psychological, and social risk factors; as a consequence, the course of both types of conditions can be more severe and complicated due to multiple persistent risk factors [14,15,16]. Additionally, the impact of substance use disorders and non-substance-use mental disorders interact, affecting the course and prognosis of both [15,17]. As a result, the management of concurrent disorders is quite complex.

The traditional approach in healthcare systems has been, and still is to address each issue separately, with limited or no standards to simultaneously address both components of concurrent disorder within the same care team. Traditional treatment methods of sequential or uncoordinated parallel care are nowadays considered obsolete. Despite new coordinated and integrated treatment approaches constituting the current standard, the majority of healthcare systems have yet to adapt.

There are still many barriers to the management and delivery of services for concurrent disorder [18,19,20,21,22]. In Canada for example, models for service delivery evolved unevenly, coordination and integration of care were limited by challenges related to the implementation of collaborative care and the need for local networks to foster service coordination and policy accountability [23,24].

The last 20 years have seen some developments, with the creation of new journals (e.g., the Journal of Dual Diagnosis) and new societies (e.g., the World Association of Dual Diagnosis). While the need for improved care for concurrent disorders is clear, the process of adapting the healthcare system to efficiently care for these individuals seems to have been slow. Clinical management guidelines are an important tool, developed to help facilitate evidence-based treatment practice.

Our purpose was to systematically review the most current clinical management guidelines for concurrent disorders and explore their scope, approach, structure, knowledge limitations, and consistency, in order to make suggestions for the future.

It is important to understand the scope of the guidelines and what they address: issues and populations. The target primary and secondary audiences may include: patients living with concurrent disorders, pharmacists, and other healthcare professionals who manage these conditions. In addition, methodological issues and issues with potential bias such as funding, the role in the design or conduct of the study, collection, analysis, and interpretation of the data or preparation, review, or approval of the guideline will be addressed.

## 2. Materials and Methods

The protocol for this systematic review was prepared according to the PRISMA-P checklist [25,26]. The review was registered in the international register—PROSPERO (International Prospective Register of Ongoing Systematic Reviews, http://www.crd.york.ac.uk/prospero).

To identify relevant guidelines, literature searches were carried out by two independent reviewers: S.H. and S.V. (in case of disagreement S.L.C. was involved and, if any discrepancy, C.S. advised) in the following electronic databases and websites: MEDLINE (via Ovid), EMBASE (via Ovid), PsycINFO, CINAHL, Trip, JouleCMA, DynaMed, SIGN, UpToDate, NICE Guidelines, and CADTH. All reviewers had completed medical training and had experience in working with individuals with concurrent disorders. Additionally, a web search for other gray literature and relevant reference lists was done. Researchers and clinicians in the field were also contacted to provide any known information about the available guidelines. All the searches were set between 1 January 2000 and 18 March 2020. Samples of keywords/MESH terms are attached in Appendix A. The inclusion criteria were to consider all published or unpublished English language formal clinical management guidelines of concurrent disorders for the appraisal of guidelines with the AGREE II (Appraisal of Guidelines for REsearch and Evaluation) tool. The AGREE (Appraisal of Guidelines for REsearch and Evaluation) instrument was developed to address the issue of variability in guideline quality, which assesses the methodological rigor and transparency of guideline development. The original AGREE tool has been refined to AGREE II [27]. In addition, guidelines addressed to all relevant professionals, patients, and their families were considered for review, but not appraised with the AGREE II. For the purpose of this work, intellectual/developmental disabilities occurring simultaneously with mental health concerns, described as “dual diagnosis” or “concurrent disorder”, were not considered. Accordingly, the exclusion criteria were: reviews of concurrent disorder management, non-English guidelines, literature addressing persons with neurodevelopmental disorders, and literature published earlier than 1 January 2000.

The search conducted revealed a total of 8841 results, comprising an electronic database search and a gray literature search. There were 8041 results from the electronic database search, which were all imported to RefWorks. After duplicate deletion, 6420 results remained. The results of the gray literature and website search (in total 800 results) were not uploaded to the RefWorks. Whenever possible, the removal of duplicate results was done manually and assessed with the same approach. From both sources, the electronic database and gray literature searches, the study titles, abstracts, and full papers were examined by both authors (S.H. and S.V.) to identify eligible studies based on the inclusion criteria. Decisions of the two authors were recorded separately and in case of disagreement, were discussed. In the absence of consensus, a decision was made by the third reviewer (S.L.C.), and finally, by the supervisory author (C.G.S.). All titles were scanned (8841) and if relevant to concurrent disorders, abstracts were read (275), and were classified for inclusion to appraise into YES, MAYBE, and NO groups. Electronic database search results were manually sorted within RefWorks, while gray literature results were manually sorted outside of it. In the YES and MAYBE groups, 75 full papers (55 from an electronic database + 20 * from gray literature) were read. A full-text review was performed for the 75 selected studies and recorded into a study selection form, documenting the reason for the exclusion and inclusion of each study. After this process, 55 papers remained that fulfilled inclusion criteria and were considered for the qualitative analysis. After full assessment, 24 papers fully fulfilled the inclusion criteria and were included in the final analysis (Figure 1: PRISMA Flow Diagram 1, Table 1). The AGREE II instrument was used to report the guidelines.

## 3. Results

In total, 24 clinical guidelines developed for concurrent disorders were included in the final analysis for appraisal by AGREE II (Table 1).

There were four Australian, one Brazilian, four Canadian, three UK, four EU, two New Zealand (one joint with Australia), five American, and one collaborative guidelines. The search yielded many different forms of information resources to manage concurrent disorders, but they were not included in this study to be appraised by AGREE II, as they were not formal clinical management guidelines. However, some of them were very comprehensive on concurrent disorder management information [1]. In addition, guidelines that were not addressed to physicians but for counselors [52] and those that were addressed to the patients and families [53,54,55], were not included. The Scottish National guideline on schizophrenia addressed concurrent disorders management only briefly, and therefore was not included in the appraisal list [56]. Similarly, toolkit [30], handbooks [57,58,59,60,61,62], reviews of current literature [19,63,64,65,66], reviews of recommendations [67,68], or adopted summaries of other guidelines [69] were not considered for inclusion within the appraisal. Lastly, some of the papers that provided concurrent disorders management related information were not included because they were only consensus recommendations for the standard of care development and suggestions for service delivery implementation [46,70,71,72,73,74,75,76,77,78,79,80,81,82,83,84,85,86,87].

Overall quality according to the AGREE II for the majority of guidelines was average (Table 2 and Table 3). Only four of the guidelines were of low quality and rated low with the AGREE II appraisal. Almost all guidelines clearly described their scope and purpose in great detail. Stakeholder involvement from different groups representing the range of views and preferences of all target groups were not considered by approximately half of the guidelines. A concern was that almost half of the guidelines showed some weaknesses in the rigor needed to comply with the standards required for developing evidence-based guidelines. Guidelines should be revised regularly to provide up to date support, however information regarding guideline updates was regularly missing.

Guidelines need to be clear and make the most important information easily identifiable. While most guidelines were clear about the recommendations, emphasis on key recommendations was often absent.

Applicability constituted perhaps the weakest domain with the most deficiencies in most guidelines: issues such as resource implications were almost never discussed, neither were issues of monitoring and/or auditing.

Lastly, information on editorial independence was missing or not clearly defined in many of the guidelines. Half of the guidelines provided no information recording this important aspect of guideline development, with failure to address competing interests of the guideline development group members.

The four Australian guidelines were all developed by the Australian Government Health Departments. All of them comprehensively covered all the questions concerning scope, purpose, and stakeholder involvement. One of the guidelines was specifically developed for primary care workers. However, none of the guidelines were developed with the maximum possible rigor. In some circumstances, information was not presented as clearly as needed for clinical practice. The implication for the resources of applying the recommendations was limited. Lastly, none of the guidelines had sufficient information on editorial independence.

Guidelines from the Brazilian Association of Studies on Alcohol and Other Drugs (ABEAD) for diagnosis and treatment of psychiatric comorbidity with alcohol and other substance dependence described clearly the scope and purpose of the guideline. However, classic guideline components including grading the evidence level and key recommendations were not mentioned. Assessment of this guideline using the AGREE II standards showed that for all domains, the information could be presented in a clearer format.

Canadian guidelines were developed by Health Canada, CANMAT, and included adapted guidelines based on UK parent guidelines. All clearly described their scope and purpose and involved stakeholders from the relevant fields in the process. However, all other domains showed room for improvement. Canadian Schizophrenia Guidelines: “Schizophrenia and Other Psychotic Disorders with Coexisting Substance Use Disorders” developed for people with schizophrenia and other psychotic disorders with coexisting substance use disorders was appraised, receiving nearly maximum scores in all domains. However, it was addressing only schizophrenia and other psychotic disorders.

UK guidelines were developed by NICE (National Institute for Health and Care for Excellence) and BAP (The British Association for Psychopharmacology). Both NICE guidelines comprehensively covered all the aspects of guideline development and scored the maximum. The guideline NG58 “Coexisting Severe Mental Illness and Substance Misuse: Community Health and Social Care Services”, was addressed to and developed for community health and social care services, and was also included in an assessment with the AGREE II, as it was recommended to read in conjunction with NICE CG 120 Clinical Guideline “Coexisting Severe Mental Illness (Psychosis) and Substance Misuse: Assessment and Management in Healthcare Settings”. The guideline NG 58 was not directly addressed to clinicians and was for the wider health and social care needs, such as employment and housing. However, both these guidelines covered different biopsychosocial aspects of concurrent disorders management with the same approach and therefore appraising them together was appropriate. Both NICE guidelines scored the highest possible with AGREE II. “BAP Updated Guidelines: Evidence-Based Guidelines for the Pharmacological Management of Substance Abuse, Harmful Use, Addiction and Comorbidity: Recommendations from BAP” were extremely clear on the scope and were developed with the utmost rigor and scored close to the possible maximum, with only minimal missing information.

The four European guidelines had very different scopes. They were developed rigorously in all domains according to the AGREE II tool assessment, however they could all be improved with clarification. “Psychiatric Comorbidity in Alcohol Use Disorders: Results from The German S3 Guidelines” developed by The German Association for Psychiatry, Psychotherapy, and Psychosomatics (DGPPN) and The German Association for Addiction Research and Therapy (DG-Sucht) for people with psychiatric comorbidity in alcohol use disorders, were the best-scored guidelines, with applicability being the main domain requiring significant improvement.

The two New Zealand guidelines had findings similar to the Australian guidelines with some domains requiring improvements. One of the guidelines was created in partnership with the Australian Government.

There were five guidelines developed in the USA. With a very different scope, they had similar overall rigor of development in all domains. However, not all clearly described information on editorial independence.

The collaborative guideline created by different stakeholders, provided a very clear scope. However, all other domains of information could be improved with minor additional information.

## 4. Discussion

This review collected all concurrent disorder English language guidelines developed over the last 20 years. Ten guidelines were developed between 2000 and 2010 and 14 between 2011 and 2020. Eight of the 14 were developed in the last five years, suggesting an increasing trend or recognition of importance. All guidelines struggled with a limited evidence base, as the pool of evidence showed limited expansion.

All guidelines were ICD/DSM based. They generally discussed specific combinations of disorders, often differentiating illicit substances and alcohol use disorders. This differentiation is consistent with established treatment providing agencies. The focus of most guidelines was on combining evidence-based interventions targeting substance use disorders with evidence-based interventions targeting non-substance-related mental disorders. Some guidelines included tobacco use disorders, while others did not. Gambling, which only recently has become part of the substance use and addiction section of the DSM and ICD, was generally not included.

Aside from the specific combination of disorders, there was little additional conceptualization of concurrent disorders. Attempts to develop psychopathological approaches that go beyond the count of symptoms are still in its infancy: e.g., the HiTOP model [88], played no role in the current conceptualization of concurrent disorders and played no role in the development of the guidelines.

There have been some attempts to develop specific models of concurrent or multimorbidity interventions, such as “patient-centered medical homes” or “Assertive Community Teams”. These attempts were sometimes mentioned but have not been considered in the guidelines as of yet. Similarly, attempts to classify approaches and levels of integration of services such as the “Levels of Collaboration. Mental Health/Primary Care Integration Options” developed by ACCT (Addiction Technology Center Transfer Network), also seem to have not become standardized enough to be utilized in guidelines [89]. None of the approaches to develop and operationalize different levels of integration have become standard enough to be included.

The level of organization of integration of care seems to not have moved beyond very basic recommendations, such as sequential, parallel, and integrated models. The sequential model suggests treating one condition, then the other. The parallel model suggests receiving mental health treatment from mental health services plus separately receiving addiction treatment from addiction services. Integrated treatment models offer one team providing mental health and addiction services within the same setting. Current evidence seems to suggest that the sequential model is obsolete, while the integrated treatment models may provide the best outcomes for the management of concurrent disorders [90]. A recent systematic review revealed that integrated models of care are more effective than conventional, nonintegrated models. Integrated models demonstrated superiority to standard care models through reductions in substance use disorders and improvement of mental health in patients with concurrent disorders. The review revealed similar findings to other studies, which indicated that the integrated model is more cost-effective than standard care [91]. Addressing both issues in an integrated manner may help to achieve better outcomes. All guidelines promoted the benefits of integrated, however, with different levels of details.

Similar in terms of simplicity and intuitiveness to the characterization of the sequential/parallel and integrated approaches is the four-quadrant framework for concurrent disorders. The four-quadrant framework has been developed to address the variability of concurrent disorders. Being a spectrum of disorders ranging from high prevalence with low impact, to low prevalence with high impact, results in considerable variation. As a result, this framework provides a model of substantial diversity in the individual treatment needs of the various people who experience concurrent disorders [92,93]. The four-quadrant model was mentioned in two guidelines developed by the U.S. Department of Health and Human Services, but played no role in specific guideline recommendations.

In order to address the level of care needs, such as indicated in a simple fashion in the four-quadrant model, an evidence-based approach to assessing severity, complexity, and need of care would be necessary. This can be in the form of staging, which has been recommended for individual disorders, but not for concurrent disorders. For example, staging has been recommended for the development of more targeted specific treatments in primary, secondary, and tertiary care settings. As concurrent disorders are closely related to severity and complexity, staging may be an issue of specific interest to concurrent disorders. However, none of the guidelines discussed or introduced staging or any similar form of determining specific levels of care.

## 5. Conclusions

Overall, specific evidence for the management of concurrent disorders continues to be rare, making it necessary for guidelines to often rely on combining evidence for individual disorders. Some studies in concurrent disorder patients indicate that certain approaches working in individual disorders are less or not effective in concurrent disorders, such as SSRIs in alcohol-dependent individuals with major depressive disorder. There is also some evidence that some medication may work better, such as clozapine for individuals suffering from schizophrenia and substance use disorder.

As current evidence suggests that better outcomes of concurrent disorder management can be achieved with integrated management approaches, broader application appears warranted. However, integrated approaches in current medical systems are rare. Furthermore, it seems that higher functionality in patients appears to allow for less integration of treatment for different disorders. Guidelines rarely allow for graded approaches and generally lack any recommendations regarding grading or staging.

Based on available evidence of this review of current guidelines quality, some of the subsections in practically all guidelines can be improved. Furthermore, certain important aspects that are essential for treatment planning are not addressed by any guideline, including the specifics of a concurrent disorder framework, the “matching” of treatment needs, and the evaluation or “staging” of the severity.

## Figures and Tables

**Figure 1 jcm-09-02406-f001:**
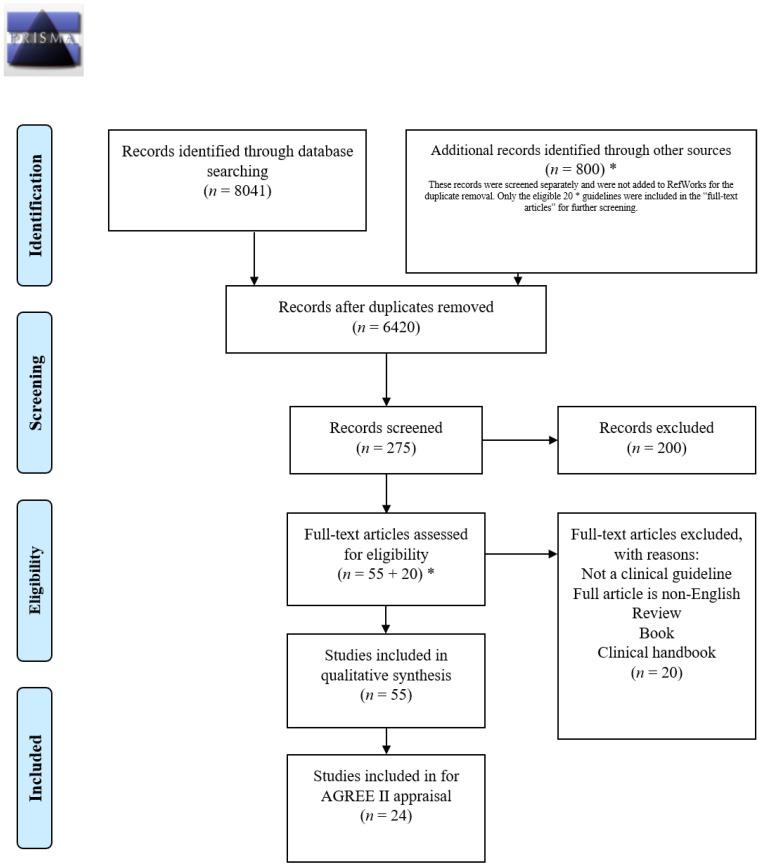
PRISMA 2009. Flow Diagram 1.

**Table 1 jcm-09-02406-t001:** Concurrent disorder guidelines included for the appraisal with the AGREE II (Appraisal of Guidelines for Research and Evaluation) tool (last reviewed 18 March 2020).

	Title	Developed by	Country	Year	Population Targeted
1	Comorbidity of Mental Disorders and Substance Use: A Brief Guide for The Primary Care Clinician [28]	Australian Government. Drug and Alcohol Services South Australia	Australia	2008	People with mental disorders and substance use
2	NSW Clinical Guidelines for the Care of Persons with Comorbid Mental Illness and Substance Use Disorders in Acute Care Settings [29]	NSW Department of Health	Australia	2009	People with comorbid mental health and substance use disorders
3	Queensland Health Dual Diagnosis Clinical Guidelines. Co-Occurring Mental Health and Alcohol and Other Drug Problems [30]	Queensland Health	Australia	2010	People with comorbid mental health and substance use disorders
4	Guidelines on The Management of Co-Occurring Alcohol and Other Drug and Mental Health Conditions in Alcohol and Other Drug Treatment Settings [31]	NHMRC Centre of Research Excellence in Mental Health and Substance Use National Drug and Alcohol Research Centre University of New South Wales	Australia	2016	Patients with alcohol and other drugs dependence and mental health conditions
5	Guidelines of The Brazilian Association of Studies on Alcohol and Other Drugs (ABEAD) for Diagnosis and Treatment of Psychiatric Comorbidity with Alcohol and Other Substance and Dependence [32]	Brazilian Association of Studies on Alcohol and Other Drugs	Brazil	2017	Alcohol and drug-dependent patients suffering from comorbid psychiatric disorders
6	Best Practices Concurrent Mental Health and Substance Use Disorders [33]	CAMH	Canada	2002	People with mental health with a substance use disorder
7	The Canadian Network for Mood and Anxiety Treatments (CANMAT) Task Force Recommendations for the Management of Patients with Mood Disorders and Comorbid Substance Use Disorders [34]	CANMAT	Canada	2012	People with mood disorders and comorbid substance use disorders
8	Concurrent Disorders Guidelines. A Supplement to The Provincial Addictions Treatment Standards [35]	Regional Health Authorities of Newfoundland and Labrador	Canada	2015	People with concurrent disorders
9	Canadian Schizophrenia Guidelines: Schizophrenia and Other Psychotic Disorders with Coexisting Substance Use Disorders [36]	CPA	Canada	2017	People with schizophrenia and other psychotic disorders with coexisting substance use disorders
10	Coexisting Severe Mental Illness (Psychosis) and Substance Misuse: Assessment and Management in Healthcare Settings [37]	NICE	UK	2011	People with coexisting severe mental illness (psychosis) and substance misuse
11	BAP Updated Guidelines: Evidence-Based Guidelines for the Pharmacological Management of Substance Abuse, Harmful Use, Addiction, and Comorbidity: Recommendations from BAP [38]	BAP	UK	2012	People with substance abuse, harmful use, addiction, and comorbidity
12	Coexisting Severe Mental Illness and Substance Misuse: Community Health and Social Care Services [39]	NICE	UK	2016	People with coexisting severe mental illness and substance misuse
13	EPA Guidance on Tobacco Dependence and Strategies for Smoking Cessation in People with Mental Illness [40]	EPA	EU	2013	People with tobacco dependence and mental illness
14	Guideline for Screening, Diagnosis, and Treatment of ADHD in Adults with Substance Use Disorders [41]	Belgian Universities and Hospital Collaborators	Belgium	2017	People with attention deficit hyperactivity disorder with substance use disorders
15	Dual Diagnosis: An Integrated Approach to Treatment: Evidence-Based Clinical Practice Guidelines [42]	Andalusian Health System Hospital	Spain	2019	People with dual diagnosis
16	Psychiatric Comorbidity in Alcohol Use Disorders: Results from The German S3 Guidelines [43]	German Association for Psychiatry, Psychotherapy, and Psychosomatics (DGPPN) and the German Association for Addiction Research and Therapy (DG-Sucht)	Germany	2017	People with psychiatric comorbidity in alcohol use disorders
17	The Assessment and Management of People with Coexisting Mental Health and Substance Use Problems [44]	New Zealand Ministry of Health	New Zealand	2010	People with coexisting mental health and substance use problems
18	Royal Australian and New Zealand College of Psychiatrists Clinical Practice Guidelines for the Management of Schizophrenia and Related Disorders [45]	Royal Australian and New Zealand College of Psychiatrists	New Zealand and Australia	2016	People with schizophrenia and related disorders
19	Improving the Care of Individuals with Schizophrenia and Substance Use Disorders: Consensus Recommendations [46]	Consensus Meeting	USA	2005	Individuals with schizophrenia and substance use disorders
20	Substance Abuse Treatment for Persons with Co-Occurring Disorders [47]	U.S. Department of Health and Human Services Substance Abuse and Mental Health Services Administration Center for Substance Abuse Treatment	USA	2005	People with co-occurring disorders
21	Substance Abuse: Clinical Issues in Intensive Outpatient Treatment [48]	U.S. Department of Health and Human Services Substance Abuse and Mental Health Services Administration Center for Substance Abuse Treatment	USA	2005	People with co-occurring disorders
22	Co-Occurring Posttraumatic Stress Disorder and Substance Use Disorder: Recommendations for Management and Implementation in the Department of Veterans Affairs [49]	Department of Veterans Affairs	USA	2011	People with co-occurring posttraumatic stress disorder and substance use disorder
23	Treatment of PTSD and Comorbid Disorders [50]	International Society for Traumatic Stress Studies	USA	2009	People with posttraumatic stress disorder and comorbid disorders
24	World Federation of Societies of Biological Psychiatry (WFSBP) Guidelines for Biological Treatment of Schizophrenia Part 3: Update. 2015. Management of Special Circumstances: Depression, Suicidality, Substance Use Disorders and Pregnancy and Lactation [51]	World Federation of Societies of Biological Psychiatry (WFSBP)	Collaboration of different countries	2015	People with schizophrenia and substance use disorders

**Table 2 jcm-09-02406-t002:** Full version of the AGREE II instrument (Strongly Disagree—1, Strongly Agree—7).

GUIDELINES (Please See Table 1: Included Guidelines)	1	2	3	4	5	6	7	8	9	10	11	12	13	14	15	16	17	18	19	20	21	22	23	24
**DOMAIN 1. SCOPE AND PURPOSE**																								
1. The overall objective(s) of the guideline is (are) specifically described.	6	7	7	6	5	7	6	7	7	7	7	7	6	7	6	7	6	7	7	7	7	7	6	7
2. The health question(s) covered by the guideline is (are) specifically described.	6	7	7	6	5	7	6	7	7	7	7	7	6	7	4	6	2	7	5	7	5	6	5	7
3. The population (patients, public, etc.) to whom the guideline is meant to apply is specifically described.	6	7	6	7	5	7	6	7	6	7	7	7	5	7	5	7	6	7	7	6	6	7	6	7
**DOMAIN 2. STAKEHOLDER INVOLVEMENT**																								
4. The guideline development group includes individuals from all relevant professional groups.	6	6	6	6	3	7	5	6	6	7	6	7	5	3	1	7	4	7	3	7	7	2	3	6
5. The views and preferences of the target population (patients, public, etc.) have been sought.	3	6	4	7	2	7	5	5	7	7	5	7	3	3	1	6	3	6	1	2	7	1	2	3
6. The target users of the guideline are clearly defined.	6	7	5	7	3	7	5	6	5	7	7	7	6	7	5	6	4	7	7	6	6	7	7	7
**DOMAIN 3. RIGOR OF DEVELOPMENT**																								
7. Systematic methods were used to search for evidence.	4	6	4	5	3	5	6	6	6	7	7	7	7	7	3	7	3	7	3	6	2	5	3	7
8. The criteria for selecting the evidence are clearly described.	7	6	4	6	2	5	6	3	5	7	7	7	6	7	3	6	5	6	1	5	5	6	1	7
9. The strength and limitations of the body of evidence are clearly described.	6	5	4	5	1	4	6	2	6	7	7	7	6	6	2	5	4	5	2	5	6	7	6	7
10. The methods for formulating the recommendations are clearly described.	5	6	4	3	1	5	6	2	6	7	7	7	5	7	1	6	4	7	4	5	5	5	5	7
11. The health benefits, side effects, and risks have been considered in formulating the recommendations.	6	7	6	2	3	5	6	5	6	7	7	7	5	5	3	5	2	7	5	3	2	3	2	6
12. There is an explicit link between the recommendations and the supporting evidence.	5	4	4	2	1	4	6	3	4	7	5	7	5	6	1	7	1	7	2	2	2	7	7	7
13. The guideline has been externally reviewed by experts prior to its publication.	2	4	2	1	1	3	3	2	7	7	5	7	4	6	3	4	1	7	6	4	4	5	1	5
14. A procedure for updating the guideline is provided.	5	6	1	7	1	3	2	2	5	7	7	7	4	2	1	2	1	2	1	2	2	7	1	2
**DOMAIN 4. CLARITY OF PRESENTATION**																								
15. The recommendations are specific and unambiguous.	5	5	4	4	3	5	4	3	6	7	6	7	6	7	3	7	6	7	7	4	5	5	5	7
16. The different options for management of the condition or health issue are clearly presented.	6	5	6	7	5	5	5	3	6	7	6	7	6	6	2	6	4	7	5	2	5	2	5	6
17. Key recommendations are easily identifiable.	3	6	6	5	1	6	2	2	6	7	6	7	4	4	6	6	7	7	7	5	6	4	6	7
**DOMAIN 5. APPLICABILITY**																								
18. The guideline describes facilitators and barriers to its application.	6	5	4	7	2	4	4	3	5	7	7	7	4	5	2	3	5	3	7	5	5	6	2	2
19. The guideline provides advice and/or tools on how the recommendations can be put into practice.	7	6	6	3	2	4	4	5	6	7	6	7	4	5	2	3	5	4	7	5	6	6	2	2
20. The potential resource implications of applying the recommendations have been considered.	3	4	5	4	3	4	3	3	5	7	5	7	4	2	1	3	4	4	5	5	5	6	1	1
21. The guideline presents monitoring and/or auditing criteria.	2	3	2	2	1	3	2	2	3	7	3	7	3	1	1	2	2	2	2	1	2	5	1	1
**DOMAIN 6. EDITORIAL INDEPENDENCE**																								
22. The views of the funding body have not influenced the content of the guideline.	4	4	2	4	4	4	4	3	6	7	3	7	4	4	3	6	2	5	3	6	6	3	2	6
23. Competing interests of guideline development group members have been recorded and addressed.	1	1	1	1	7	1	6	2	7	7	6	7	6	2	1	7	1	7	1	5	6	1	1	7
1. Rate the overall quality of this guideline. Lowest possible quality—1 Highest possible quality—7	5	5	5	5	3	5	5	4	6	7	6	7	5	5	3	6	4	6	4	5	5	5	3	6
OVERALL CALCULATED BY DOMAIN AVERAGE	5	5	5	5	4	5	5	4	6	7	6	7	5	5	3	6	3	4	3	4	4	4	3	4
2. I would recommend this guideline for use. Yes—1, Yes with Modifications—2, No—3	2	2	2	2	2	2	2	2	2	1	2	1	2	2	2	2	2	2	2	2	2	2	2	2
NOTES																								

**Table 3 jcm-09-02406-t003:** Short version of the Agree II instrument (Strongly Disagree—1, Strongly Agree—7).

GUIDELINES (Please See Table 1: Included Guidelines)	1	2	3	4	5	6	7	8	9	10	11	12	13	14	15	16	17	18	19	20	21	22	23	24
**DOMAIN 1. SCOPE AND PURPOSE**	6	7	7	6	5	7	6	7	7	7	7	7	6	7	5	7	5	7	6	7	6	7	6	7
**DOMAIN 2. STAKEHOLDER INVOLVEMENT**	5	6	5	7	3	7	5	6	6	7	6	7	4	4	2	6	4	7	4	5	7	3	4	5
**DOMAIN 3. RIGOR OF DEVELOPMENT**	5	6	4	4	2	4	5	3	6	7	7	7	5	6	2	5	3	5	3	4	4	6	3	6
**DOMAIN 4. CLARITY OF PRESENTATION**	5	5	5	5	3	5	4	3	6	7	6	7	5	6	4	6	6	7	6	4	5	4	5	7
**DOMAIN 5. APPLICABILITY**	5	5	4	4	2	4	3	3	5	7	5	7	4	3	2	3	4	3	5	4	5	6	2	2
**DOMAIN 6. EDITORIAL INDEPENDENCE**	3	3	2	3	6	3	5	3	7	7	5	7	5	3	2	7	2	6	2	6	6	2	2	7
1. Rate the overall quality of this guideline.Lowest possible quality—1Highest possible quality—7	5	5	5	5	4	5	5	4	6	7	6	7	5	5	3	6	3	4	3	4	4	4	3	4
2. I would recommend this guideline for use.Yes—1, Yes with Modifications—2, No—3	2	2	2	2	2	2	2	2	2	1	2	1	2	2	2	2	2	2	2	2	2	2	2	2
NOTES

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
