# Peer review of "Concurrent Disorder Management Guidelines. Systematic Review"

_jcm, 2020, doi:10.3390/jcm9082406_

Round 1

Reviewer 1 Report

This review did a very thorough job of examining existing guidelines from across the world that concern the treatment of adult concurrent disorders. I appreciate the careful work that went into identifying the guidelines to include in this review and use of the AGREE II tool to evaluate each of these guidelines. It is an added strength of this manuscript that the review was registered in PROSPERO. Below, are some point-by-point critiques, which if addressed, would further strengthen this manuscript.

  1. The sentence on lines 69-71, beginning with “Gaps between need for…” is unclear. Please consider revising.
  2. Please include a include a brief description of AGREE II in the introduction or methods for readers who may not be familiar with this tool.
  3. In the text of the results section, it would be helpful to reference the tables in relation to where the text describes the results of the table (versus referencing all of the tables within the first sentence of the results). For instance, authors could reference Table 1 where they describe the guidelines that were reviewed. Tables 2 and 3 could be referenced where the quality of the guidelines is discussed.
  4. I found myself getting a bit lost in Table 2. It might be helpful to repeat the heading for every new page the table falls on (though I acknowledge that formatting will likely be altered when publication goes to print).
  5. It would be helpful to include in the results section which guidelines promote which models of care (e.g., sequential, parallel, integrated) and which at least make reference to the four-quadrant framework.

Author Response

  1. The sentence on lines 69-71, beginning with "Gaps between need for..." is revised to make it clear.
  2. A description of AGREE II is added for clarity.
  3. Tables are referenced as advised in relation to where the text describes the results of table.
  4. I could not add headings on each new page for tables because of the formatting changes were done. I will contact that they can allow me to do those changes.
  5. Detailed descriptions of which guidelines promote different models of care were added

Reviewer 2 Report

Thank you for this article. It is well written, balanced and I enjoyed reading.

Please see below some minor suggestions:

Abstract:

Lines 18-20: the sentence doesn’t flow well. Please re-write.

Lines 20-21: please review grammar

Introduction:

Line 69: what is the main cause(s) of increasing burden of morbidity and mortality

Line 76: is Dual diagnosis has become more complicated since or due to the emergence of NPS?

Line 89: define vulnerable

Table 1: in addition to date of publishing, it would be useful to include date last reviewed

Lines 166-176 contain useful points that can be parts of the inclusion/exclusion criteria in the methodology.

Line 247: need to introduce table 2 first before presenting it. Same for all other tables.

Table 2: row 1: GUIDELINES (Please see Table of Included Guidelines). Please change to (Please see Table 1). Same for Table 3.

Author Response

Lines 18-20: the sentence is re-writen to flow.

Lines 20-21: Grammar change.

Introduction:

Line 69: the main cause(s) of increasing burden of morbidity and mortality is clarified

Line 76: is Dual diagnosis has become more complicated since or due to the emergence of NPS? PLEASE CLARIFY WHAT IS NPS?

Line 89: Vulnerable is defined

Table 1: Date last reviewed is added

Lines 166-176 points added to the inclusion/exclusion criteria in the methodology.

Line 247: Tables introduced first.

Table 2: row 1: GUIDELINES (Please see Table of Included Guidelines). Changed.